PREPARED FOR SUBMISSION TO JHEP

# SymTFTs for U(1) symmetries from descent

**Finn Gagliano and Iñaki García Etxebarria**

*Department of Mathematical Sciences, Durham University,*
*Durham, DH1 3LE, United Kingdom*

*E-mail:* finn.gagliano@durham.ac.uk,
inaki.garcia-etxebarria@durham.ac.uk

ABSTRACT: Recently, the notion of symmetry descent has been introduced in order to obtain the $(d + 1)$-dimensional Symmetry TFT (SymTFT) of a $d$-dimensional QFT from the edge mode behaviour of a theory in $(d + 2)$-dimensions. This method has so far been used to obtain SymTFTs for discrete higher-form symmetries of geometrically engineered QFTs. In this note, we extend the symmetry descent procedure to obtain SymTFTs for $U(1)$ higher-form symmetries of geometrically engineered QFTs. We find the resulting SymTFTs match those in the works of Antinucci-Benini and Brennan-Sun.

## 1 Introduction

There has been a great increase in our understanding of Symmetry TFTs (SymTFTs) since their origins in [1–5]. SymTFTs have been extensively studied in the context of finite discrete higher-form symmetries, and extensions to continuous higher-form symmetries [6–8] and non-invertible symmetries [9, 10] are known. Non-topological generalisations have been discussed in [11], and more recently SymTFTs for theories defined on a manifold with boundary have been described in [12–21]. The literature on these topics is already vast, so we have included only some initial pointers into the relevant literature, we refer the reader to [22–24] for reviews and additional references. For reviews of categorical symmetries more generally, see [25–31].

Thus far, there have been string theory derivations of SymTFTs for discrete higher-form symmetries [4, 32, 33], as well as for continuous and non-invertible symmetries described by a SymTh [11]. An outline of how SymTFTs for continuous non-abelian 0-form symmetries could be obtained from string theory was given in [8], but we will be focusing only on abelian higher-form symmetries in this work. Applications of SymTFTs from string theory to holography have also been considered [34–36], as well as interpreting the SymTFT as a theory of gravity via the holographic principle [37]. Additionally, in [38] a proposal

was given for obtaining SymTFTs for continuous abelian symmetries from M-Theory using similar methods to those proposed in [8].

In [4], SymTFTs for discrete higher-form symmetries were obtained from the boundary of the engineering geometry, with the discrete defects of the SymTFT arising from torsional fluxes on this boundary. In particular, the anomaly theory was obtained from dimensionally reducing the topological sector of M-Theory on the link of the singular locus in the geometry, and the presence of the $BF$ sector of the SymTFT had to be deduced from arguments following [39–42]. In [32], following [43–46], the $BF$ sector was obtained by considering the fields of Type II string theory as parameters of gauge transformations on the boundary of an 11-dimensional Maxwell-$BF$ theory. This approach allows for the treatment of both electric and magnetic degrees of freedom at the same time, which is crucial for determining the $BF$ action for the discrete sector of the SymTFT. The resulting "symmetry descent" procedure (which we will review in detail below) leads to the following formula for the Lagrangian $\mathcal{L}_{\mathrm{Sym}}$ of the SymTFT

$$\Delta \int_L \mathcal{L}_{\mathrm{bulk}} = \delta \mathcal{L}_{\mathrm{Sym}} \tag{1.1}$$

where the left-hand side describes the gauge variation of the dimensional reduction on the linking geometry, and the right-hand side is the derivative of the Lagrangian for the $BF$ sector of the SymTFT. The name "symmetry descent" is due to the similarity of (1.1) with the usual anomaly descent equation.

The aim of this note is to use the symmetry descent method to rederive from a geometric perspective the SymTFT for $U(1)$ higher-form symmetries given in [6, 7]. For $d$-dimensional QFTs geometrically engineered in a Type II string theory on a space $X_{10-d}$ with boundary $L_{9-d}$, our result is the following $BF$ sector of the SymTFT

$$\mathcal{L}_{BF} = 2\pi i \sum_{k,l} K_{i,j}(g_k, H_{l-1})\tilde{n}_{i-k-1} \cup \delta\tilde{m}_{j-l-1} + J_{i,j}(g_k, V_l)n_{i-k-1} \cup \delta B_{j-l-2} \tag{1.2}$$

where $K_{i,j}$ is a rational number and $J_{i,j}$ is an integer, both defined in terms of intersection numbers of $L_{9-d}$, and the $n, \tilde{n}, \tilde{m}$ fields are real cochains that integrate to an integer, and $B$ is a real cochain. The Lagrangian (1.2) describes both the finite and $U(1)$ symmetry sectors: the first term in (1.2) is the usual $BF$ sector for discrete higher-form symmetries, and the second corresponds to the SymTFT that describes $U(1)$ symmetries given in [6, 7].

This note is organized as follows. In section 2 we review background material necessary for our later discussion. In section 3 we derive the $U(1)/\mathbb{R}$ SymTFT for theories engineered on a conical geometry $X_{10-d}$ with link $\partial X_{10-d} = L_{9-d}$ using the symmetry descent procedure, and in section 4 we derive a simplified differential form version of the SymTFT by performing a standard Kaluza-Klein reduction of the bulk action on the link $L_{9-d}$, obtaining similar results to those in [38]. In section 5 we discuss the example of the 6d $\mathcal{N} = (1,1)$ and $\mathcal{N} = (2,0)$ $\mathfrak{su}(N)$ theories. The analysis of this example will bring to light various subtleties of the procedure, which we solve in section 6 by going to differential $K$-theory, and where we also discuss how to incorporate anomalies in our discussion.

## 2 Review of background material

### 2.1 Symmetry inflow and the Hopkins-Singer formalism

The bulk actions considered in [32] to obtain the SymTFT from string theory are inspired by the problem of writing down an action for a self-dual field. Consider a self-dual field strength $F_{2p+1}$ in $4p+2$ dimensions, so that $*F_{2p+1} = F_{2p+1}$. Naively, the action for such a field is

$$\int_{X_{4p+2}} F_{2p+1} \wedge *F_{2p+1} = \int_{X_{4p+2}} F_{2p+1} \wedge F_{2p+1} = 0 \,. \tag{2.1}$$

There are different ways of addressing this problem [43–45, 47–64]. We will focus on the approach initiated in [32, 43–46], in which one considers a Maxwell-$BF$ theory in $4p+3$ dimensions, on some $Y_{4p+3}$ with $\partial Y_{4p+3} = X_{4p+2}$, and treats the field strengths $F_{2p+1}$ as gauge parameters of the fields in this bulk theory

$$S_{\text{bulk}} = \int_{Y_{4p+3}} \frac{1}{2e^2} dB_{2p+1} \wedge *dB_{2p+1} + \frac{1}{2m^2} dC_{2p+1} \wedge *dC_{2p+1} + k B_{2p+1} \wedge dC_{2p+1} \,. \tag{2.2}$$

A $BF$ theory on a manifold with boundary is not invariant under the gauge transformations $B_{2p+1} \to B_{2p+1} + d\lambda_{2p}^B, C_{2p+1} \to C_{2p+1} + d\lambda_{2p}^C$ due to non-vanishing total derivatives, and it was shown in [32, 46] that these boundary terms obey Maxwell's equations on the boundary, with $d\lambda_{2p}^B \equiv F_{2p+1}, d\lambda_{2p}^C \equiv *F_{2p+1}$.

A crucial aspect of the derivation for the symmetry descent procedure introduced in [32] was the formulation of the theory in terms of Hopkins-Singer differential cochains [65]. This was necessary in order to consider the effects of torsional elements of the cohomology of the linking geometry, which result in the discrete higher-form symmetries of the SymTFT. While it is possible in principle to see the effects of free cocycles without using a differential uplift of the cohomology, we still find it useful for our purposes to keep this uplift. Introductions for physicists to the Hopkins-Singer formalism were given in [32, 46], so we will be brief in our review below. We follow the conventions in [32]. In particular, we use $\check{C}(n)^p(\mathcal{M}), \check{Z}(n)^p(\mathcal{M})$ to denote the space of (bi-graded) Hopkins-Singer differential cochains and differential cocycles respectively, and denote $\check{C}^p(\mathcal{M}) \coloneqq \check{C}(p)^p(\mathcal{M})$. We will denote a differential cochain $\check{a}_p \in \check{C}(n)^p(\mathcal{M})$ with a subscript for its degree unless stated otherwise. Such an $\check{a}_p$ is then given by

$$\check{a}_p = (N_p, A_{p-1}, F_p) \in C^p(\mathcal{M}; \mathbb{Z}) \times C^{p-1}(\mathcal{M}; \mathbb{R}) \times \Omega^p(\mathcal{M}; \mathbb{R}) \tag{2.3}$$

where $C^p(\mathcal{M}; G)$ is the space of $G$-valued $p$-dimensional cochains, and $\Omega^p(\mathcal{M}; \mathbb{R})$ is the space of differential $p$-forms. If $p < n$ we set $F_p = 0$. We can give a more familiar meaning to these terms by briefly introducing some notions recapped in [46]. All of the gauge-invariant information of a gauge field $A_p$ is encoded in the pair $(F_{p+1}, \chi)$, where $F_{p+1}$ is the field strength of the gauge field, which should be gauge-invariant, and $\chi : Z_p \to U(1)$, is a map from closed $p$-dimensional submanifolds to U(1), called the higher holonomy function

$$\chi(\Sigma_p) = e^{2\pi i \int_{\Sigma_p} A_p} \tag{2.4}$$

such that a shift $A_p \to A_p + n$ for $n \in \mathbb{Z}$ leaves the holonomy function (and the field strength) invariant. This is the Cheeger-Simons formulation of differential cohomology in terms of differential characters. The formulation in terms of the differential characters $\chi$ captures all of the gauge-invariant information of the gauge field, but we wish to discuss gauge parameters of the gauge field, as these furnish our Maxwell degrees of freedom on the boundary of our bulk theory, so we need to work at the level of the differential cochains $\check{C}^p(\mathcal{M})$. (In section 6 we will have to go one step further, and discuss differential refinements of generalized cohomology theories.) We define for convenience the following maps for a Hopkins-Singer cochain $\check{a}_{p+1}$:

$$\mathsf{h}(\check{a}_{p+1}) = A_p \in C^p(\mathcal{M}; \mathbb{R}), \tag{2.5}$$

$$I(\check{a}_{p+1}) = N_{p+1} \in C^{p+1}(\mathcal{M}; \mathbb{Z}), \tag{2.6}$$

$$R(\check{a}_{p+1}) = F_{p+1} \in \Omega^{p+1}(\mathcal{M}; \mathbb{R}). \tag{2.7}$$

Consider $[F_{p+1}] \in H^{p+1}(\mathcal{M}; \mathbb{R})$ as a (potentially trivial) cohomology class. Then we define $[N_{p+1}] \in H^{p+1}(\mathcal{M}; \mathbb{Z})$ to be the (again, potentially trivial) integral cohomology class that maps to $[F_{p+1}]$ under the natural map $H^{p+1}(\mathcal{M}; \mathbb{Z}) \to H^{p+1}(\mathcal{M}; \mathbb{R})$. This is an integral up-lift of the de Rham field strength, the characteristic class. By definition of integral cochains, $\int N_{p+1} \in \mathbb{Z}$. Note that if $[N_{p+1}] \in \text{Tor } H^{p+1}(\mathcal{M}; \mathbb{Z})$ then $[F_{p+1}] = 0 \in H^{p+1}(\mathcal{M}; \mathbb{R})$. See [40] for a nice explanation of this.

In [32, 44–46], the RR sector of the Type II string theories was then given by 11d bulk actions similar to those for self-dual gauge fields

$$S = 2\pi i \mathsf{h} \left( \int_{\mathcal{N}_{d+2} \times L_{9-d}} \check{a}_i \cdot \check{a}_j \right) \tag{2.8}$$

where $\partial \mathcal{N}_{d+2} = \mathcal{M}_{d+1}$ is the spacetime on which the SymTFT of a d-dimensional QFT $\mathcal{T}$ will be defined, $\partial X_{10-d} = L_{9-d}$ is the link of the geometric engineering geometry, where we assume for simplicity that $X_{10-d}$ is a toric variety, such that we can consider the defect group of $\mathcal{T}$ to arise from generators of the homology of $L_{9-d}$ — see [66, 67] for an explanation of this. To condense notation, we will sometimes write $Y_{11} := \mathcal{N}_{d+2} \times L_{9-d}$. Finally, $\check{a}_i, \check{a}_j$ are two differential cochains in the 11d bulk geometry such that $i + j = 12$, where we are no longer considering only self-dual fields. These cochains have gauge transformations

$$\check{a}_i \to \check{a}_i + d\check{b}_{i-1} \tag{2.9}$$

and we note that the actions in the form of (2.8) are not gauge invariant when $\partial Y_{11} \neq \varnothing$, similar to the bulk actions for self-dual fields. This is crucial for the works of [32, 46], and will continue to be for what we discuss here. In this note, we are considering only Type II string theories, and so $\check{a}_i \cdot \check{a}_j$ terms in the bulk theory are $BF$ terms that correspond to terms of the form

$$\int_{\mathcal{M}_{10}} F_{i-1} \wedge *F_{i-1} \tag{2.10}$$

of the 10d Type II actions, where $\mathcal{M}_{10} = \partial Y_{11} = \mathcal{M}_{d+1} \times L_{9-d}$. The Type II field strengths $F_{i-1}$ appear in the 11d bulk as gauge parameters of the $\check{a}_i$ fields. Thus, if we have, say,

Type IIA, then we would have terms of the form above for $F_2, F_4$, giving us the required 11d bulk action[1]

$$S = 2\pi i \mathsf{h} \left( \int_{Y_{11}} \check{a}_3 \cdot \check{a}_9 + \check{a}_5 \cdot \check{a}_7 \right) . \tag{2.11}$$

We could then do the same analysis for Type IIB to get a similar 11d action corresponding to the $F_1, F_3, F_5$ field strengths:

$$S = 2\pi i \mathsf{h} \left( \int_{Y_{11}} \check{a}_2 \cdot \check{a}_{10} + \check{a}_4 \cdot \check{a}_8 + \check{a}_6 \cdot \check{a}_6 \right) \tag{2.12}$$

where $\check{a}_6$ is dotted with itself - this comes from the fact that the $F_5$ field strength is self-dual, i.e. $F_5 = *F_5$.

As has been analysed in detail in [32, 46], one can let the $\check{a}_i$ field become pure gauge near the boundary of the bulk, i.e. we let $\check{a}_i = d\check{b}_{i-1}$, such that, near the boundary, we have the gauge non-invariance of the action in (2.8)

$$S = 2\pi i \mathsf{h} \left( \int_{Y_{11}} d\check{b}_{i-1} \cdot d\check{b}_{j-1} \right) \tag{2.13}$$

with the SymTFT Lagrangian being obtained from the symmetry descent equation

$$\mathsf{h} \left( \int_{L_{9-d}} d\check{b}_{i-1} \cdot d\check{b}_{j-1} \right) = \delta \mathcal{L}_{\mathrm{Sym}} \bmod 1. \tag{2.14}$$

The $\check{b}_{i-1}$ field corresponds to the electric field strength $F_{i-1}$ of the 10d string theory, and $\check{b}_{j-1}$ similarly corresponds to the magnetic dual field strength. See [32, 46] for an explanation of this correspondence between an 11d bulk Maxwell-$BF$ theory and the 10d Maxwell on the boundary.

Drawing from the ideas of [66, 67], we get a contribution to the defect group of the engineered $d$-dimensional QFT $\mathcal{T}$ by wrapping $Dp$-branes around non-compact $(k + 1)$-cycles in the engineering geometry $X_{10-d}$, and this corresponds to an expansion of the field strengths of these fluxes into components on the spacetime of the QFT and the engineering geometry. For example, to obtain the SymTFT for discrete higher-form symmetry corresponding to wrapping a $D(i-3)$ brane around a torsional $(k-1)$-cycle of the link $L_{9-d}$, one can make the ansatz

$$\check{b}_{i-1} = \check{\beta}_{i-k-1} \cdot \check{t}_k \tag{2.15}$$

for both $\check{b}_{i-1}, \check{b}_{j-1}$, where $\check{t}_k$ is a flat differential uplift of some torsion generator $t_k \in \mathrm{Tor} H^k(L_{9-d}; \mathbb{Z}) = \mathrm{Tor} H_{k-1}(L_{9-d})$. Then, substituting this ansatz into (2.14) gives us

$$\mathcal{L}_{\mathrm{sym}} = \frac{1}{n} \mathsf{h}(\check{\beta}_{i-k-1} \cdot \check{\beta}_{j-k-1}) \tag{2.16}$$

which is a $BF$ term of the SymTFT for a $\mathbb{Z}_n$ higher-form symmetry.

---

[1] The product of differential cochains is not (graded) commutative, so there is an ambiguity regarding the ordering of the terms in (2.11). We will address this issue explicitly below.

As $\breve{b}_{i-1}$ corresponds to an $F_{i-1}$ field strength of Type II string theory, it couples to an electric $D(i-3)$ brane, and $\breve{b}_{j-1}$ to the magnetic dual $D(j-3) = D(9-i)$ brane, where we have used that $i + j = 12$. Therefore, the electric flux associated to a $D(i-3)$ brane is

$$\int_{\Sigma_{i-1}} R(\breve{b}_{i-1}) \tag{2.17}$$

and the magnetic flux associated to the dual $D(9-i)$ brane is

$$\int_{\Sigma_{j-1}} R(\breve{b}_{j-1}) \tag{2.18}$$

Following the discussion in [50], these fluxes should really be quantized in an appropriate $K$-theory class, but for now we can naively assume that these are integral. We discuss K-theoretic aspects of the formulation in section 6, and will find that the pursuit of SymTFTs for both continuous and discrete higher-form symmetries of a theory will lead us to actions most naturally formulated in terms of differential $K$-theory.

## 2.2   SymTFTs for $U(1)$ symmetries

The SymTFT for a $U(1)$ symmetry was put forward in [6, 7]:

$$S = \int_{Y_{d+1}} b_{d-p-1} \wedge dA_{p+1} \tag{2.19}$$

where $b_{d-p-1}$ is an $\mathbb{R}$ $(d-p-1)$-form gauge field, and $A_{p+1}$ is a $U(1)$ $(p+1)$-form gauge field. The topological operators of such a theory are

$$U_\alpha = e^{i\alpha \int b_{d-p-1}}, W_q = e^{iq \int A_{p+1}} \tag{2.20}$$

where $\int b_{d-p-1} \in \mathbb{Z}$, $\alpha \in U(1), q \in \mathbb{Z}$. There are then 3 possible boundary conditions for these operators: firstly, if we pick Dirichlet for $W_q$, and Neumann for $U_\alpha$, then this corresponds to a $U(1)^{(p)}$ symmetry of the $d$-dimensional theory. If we choose instead to have Dirichlet for $U_\alpha$ and Neumann for $W_q$, then we have a $\mathbb{Z}^{(d-p-2)}$ symmetry, and the gauging of the $U(1)^{(p)}$ symmetry in this case is understood as gauging $U(1)$ with the discrete topology, sometimes called flat gauging. The final possible boundary condition is to give $W_{bq}, U_{\widetilde{\alpha}}$ Dirichlet boundary conditions, where $\widetilde{\alpha} \in U(1)/(\frac{1}{b}\mathbb{Z}) \cong U(1)$ while giving $W_k, U_{\frac{2\pi n}{b}}$ Neumann boundary conditions, where $k \in \mathbb{Z}_b, n \in \mathbb{Z}, b \in \mathbb{Z}_{>1}$. Then, letting $U_{\widetilde{\alpha}}$ link $W_{bq}$ and $W_k$ link $U_{\frac{2\pi n}{b}}$, we get a $U(1)^{(p)} \times \mathbb{Z}_b^{(d-p-2)}$ symmetry. In [7], this final boundary condition could be inferred by gauging a $\mathbb{Z}_b$ subgroup of the $U(1)^{(p)}$ symmetry such that we obtain a $\mathbb{Z}_b^{(d-p-2)}$ dual global symmetry. If we were to gauge the whole $U(1)^{(p)}$ symmetry, but now *without* discrete topology by introducing a dynamical photon to the theory, this maps the original SymTFT to another for the dual $U(1)^{(d-p-3)}$ symmetry

$$S_{\text{gauged}} = \int_{Y_{d+1}} c_{p+2} \wedge dB_{d-p-2} \tag{2.21}$$

where $B_{d-p-2}$ is a $U(1)$ gauge field corresponding to this dual $U(1)^{(d-p-3)}$ symmetry, and $c_{p+2}$ is another $\mathbb{R}$ gauge field. [7] refers to this as dynamical gauging.

## 3 SymTFTs for $U(1)$ symmetries from Type II string theory

We will now show how to extend the symmetry descent procedure to the case of continuous abelian symmetries. In section 4 we will use a Kaluza-Klein approach using differential forms rather than differential cohomology, which doesn't allow us to discuss torsion generators and thus discrete higher-form symmetries, but allows us to get a better feeling of the physics of this formulation. The main extension of [32] that leads us to terms in the SymTFT for continuous higher-form symmetries is adjusting the ansatz of (2.15): to include continuous symmetries in the symmetry descent procedure we can pick the most general ansatz possible, notably including free cocycles, not just torsion:

$$\check{b}_{i-1} = \check{\beta}_{i-1} + \sum_{k=0}^{9-d} \sum_{\alpha} \check{\beta}_{i-k-1}^{\alpha} \cdot \check{g}_k^{\alpha} \tag{3.1}$$

where $\alpha$ indexes the various generators $g_k^{\alpha} \in H^k(L_{9-d}; \mathbb{Z})$ such that $\check{g}_k^{\alpha} \in \check{Z}^k(L_{9-d})$ and $\check{\beta}_{i-k-1}^{\alpha} \in \check{C}(i-k)^{i-k-1}(\mathcal{N}_{d+2})$.[2] In this ansatz, we must only sum over the $\check{g}_k$ whose corresponding non-compact cycle in homology can have the associated $D(i-3)$-brane wrapped around it to give a defect in $\mathcal{T}$, and similar for the dual magnetic brane and the expansion of $\check{b}_{j-1}$.

Explicitly, this means that $\check{g}_k$ must be the differential uplift of the cohomology dual to a generator of $\frac{H_k(X_{10-d}, L_{9-d})}{H_k(X_{10-d})}$, but in this note we will assume for all examples the isomorphism $\frac{H_k(X_{10-d}, L_{9-d})}{H_k(X_{10-d})} = H_{k-1}(L_{9-d})$ for simplicity. The reason we require such $\check{g}_k$ is that wrapping branes around cycles in $\frac{H_k(X_{10-d}, L_{9-d})}{H_k(X_{10-d})}$ gives us the defect group $\mathbb{D}$ of our QFT $\mathcal{T}$ which then corresponds to the higher-form symmetries of our theory, and so it is appropriate to only include the corresponding generators $\check{g}_k \in \check{Z}^k(L_{9-d})$ in our ansatz. We will discuss the defect group in more detail in section 5. As we have chosen $\check{g}_k^{\alpha} \in \check{Z}^k(L_{9-d})$, we have $d\check{g}_k^{\alpha} = 0$. This is analogous to a Kaluza-Klein reduction, where picking non-closed $\check{g}_k^{\alpha}$ would give massive modes after a dimensional reduction. We are interested only in IR behaviour, where such massive modes should be integrated out, and thus we expand in terms of closed generators $\check{g}_k^{\alpha}$. For the same reason, we choose that torsion generators $\check{g}_k = \check{t}_k$ are flat to avoid any massive modes, i.e. $R(\check{t}_k) = 0$.

We then have for $\check{t}_k = (t_k, \varphi_{k-1}, 0)$

$$d\check{t}_k = (\delta t_k, -t_k - \delta\varphi_{k-1}, 0) = 0 \tag{3.2}$$

which implies

$$t_k = -\delta\varphi_{k-1} \tag{3.3}$$

Free generators $\check{g}_k = \check{f}_k$ cannot be flat, so for $\check{f}_k = (f_k, h_{k-1}, v_k)$, we have that

$$d\check{f}_k = (\delta f_k, v_k - f_k - \delta h_{k-1}, \delta v_k) = 0 \tag{3.4}$$

and therefore

$$\delta h_{k-1} = v_k - f_k \tag{3.5}$$

---

[2]Note that in (3.1) we are leaving implicit the pullbacks of the $\check{\beta}, \check{g}$ to $Y_{11}$, and will continue to do so throughout.

A useful corollary of this is

$$\int_{\Sigma_k} v_k = \int_{\Sigma_k} f_k. \tag{3.6}$$

for closed $k$-dimensional submanifolds $\Sigma_k$ of $L_{9-d}$. We can see this using Stokes' theorem

$$\int_{\Sigma_k} v_k - f_k = \int_{\Sigma_k} \delta h_{k-1} = \int_{\partial \Sigma_k = \varnothing} h_{k-1} = 0. \tag{3.7}$$

Getting back to the ansatz in (3.1), we have from [46] that for $\check{b}_{i-1}$ to be valid gauge parameters, we require the $\check{a}_i$ fields to be left invariant under the identification

$$\check{a}_i = (I_i, A_{i-1}, F_i) \sim (I_i - \delta M_{i-1}, A_{i-1} + M_{i-1} - \delta \alpha_{i-2}, F_i). \tag{3.8}$$

In [32], the gauge parameters $\check{b}_{i-1}$ were given as

$$\check{b}_{i-1} = (f_{i-1}, \lambda_{i-2}, 0), \tag{3.9}$$

i.e. $\check{b}_{i-1} \in \check{C}(i)^{i-1}$, which is a flat field, and is the space of gauge parameters for our fields.

There is one last thing that we would like to emphasize before moving onto the symmetry descent procedure. Suppose we have some $\check{b}_i = (f_i, \lambda_{i-1}, 0)$, then its derivative is given as

$$d\check{b}_i = (\delta f_i, -f_i^{\mathbb{R}} - \delta \lambda_{i-1}, 0). \tag{3.10}$$

Here we have written $\mathbb{R}$ superscripts for the $f_i$ term in $\mathsf{h}(d\check{b}_i)$ to emphasize that we have promoted $f_i \in C^i(\mathcal{M}; \mathbb{Z})$ to $f_i^{\mathbb{R}} \in C^i(\mathcal{M}; \mathbb{R})$ via the natural inclusion map. More on this can be found in [46, 50]. Thus, we are almost thinking of $f_i^{\mathbb{R}}$ as an $\mathbb{R}$ gauge field that integrates to an integer. This subtle detail has some importance for our final result, but from now on we will leave the superscripts implicit unless directly relevant, as is often done in the literature.

We would now like to use the ansatz in (3.1) to perform symmetry descent. From the discussion above, we have that the $\check{\beta}_{i-k-1}$ from our ansatz in (3.1) are of the form

$$\check{\beta}_{i-k-1} = (n_{i-k-1}, A_{i-k-2}, 0) \tag{3.11}$$

so that

$$d\check{b}_{i-1} = d\check{\beta}_{i-1} + \sum_k \sum_\alpha (d\check{\beta}_{i-k-1}^\alpha) \cdot \check{g}_k^\alpha \tag{3.12}$$

as $d\check{g}_k^\alpha = 0$, where

$$d\check{\beta}_{i-k-1} = (\delta n_{i-k-1}, -n_{i-k-1} - \delta A_{i-k-2}, 0) \tag{3.13}$$

such that, leaving $\alpha$ indices implicit and writing $\check{g}_k = (g_k, h_{k-1}, v_k)$ with $v_k = 0$ if $g_k$ is torsion,

$$d\check{b}_{i-1} = \sum_{k,\alpha} (\delta n_{i-k-1} \cup g_k, (-1)^{i-k} \delta n_{i-k-1} \cup h_{k-1} - (n_{i-k-1} + \delta A_{i-k-2}) \cup v_k, 0). \tag{3.14}$$

Similarly, we write $\check{\beta}_{j-l-1}$, the second gauge parameter terms

$$\check{\beta}_{j-l-1} = (m_{j-l-1}, B_{j-l-2}, 0) \tag{3.15}$$

where the generator it couples to is written as

$$\check{G}_l^\gamma = (G_l^\gamma, H_{l-1}^\gamma, V_l^\gamma) \tag{3.16}$$

where the $\gamma$ indices label each generator of $\check{Z}^l(L_{9-d})$, and we will leave these indices implicit from now on. Now, temporarily leaving the subscript indices indicating the degree of the cochains and forms implicit also, we have

$$d\check{b}_{j-1} = \sum_{\gamma,l}(\delta m \cup G, (-1)^{j-l}\delta m \cup H - (m + \delta B) \cup V, 0). \tag{3.17}$$

The product of these two gauge parameters is then given by

$$d\check{b}_{i-1} \cdot d\check{b}_{j-1} =$$
$$\sum_{\alpha,\gamma,k,l} (\delta n \cup g \cup \delta m \cup G, (-1)^i \delta n \cup g \cup \left[(-1)^{j-l}\delta m \cup H - (m + \delta B) \cup V\right], 0). \tag{3.18}$$

We are interested in the integral on the left-hand side of (2.14), which we can write now as

$$\sum_{\alpha,\gamma,k,l} \int_{L_{9-d}} (-1)^i \delta n \cup g \cup \left[(-1)^{j-l}\delta m \cup H - (m + \delta B) \cup V\right] \tag{3.19}$$

where we are still leaving indices and subscripts implicit. Then, this integral can be written as

$$\sum_{\alpha,\gamma,k,l} K_{i,j}(g, H)\delta n_{i-k-1} \cup \delta m_{j-l-1} + J_{i,j}(g, V)\delta n_{i-k-1} \cup (\delta B_{j-l-2} + m_{j-l-1}) \tag{3.20}$$

where we define

$$K_{i,j}(g_k, H_{l-1}) := (-1)^{i+j-l+k(j-l)} \int_{L_{9-d}} g_k \cup H_{l-1}, \tag{3.21}$$

$$J_{i,j}(g_k, V_l) := (-1)^{i+1+k(j-l-1)} \int_{L_{9-d}} g_k \cup V_l. \tag{3.22}$$

These integrals are dependent on a number of factors, and so we have extracted them here to simplify our result. An important note is that $J_{i,j}$ is always an integer: if $R(\check{G}_l) = V_l$ where $\check{G}_l$ is the differential refinement of a torsion generator, then $V_l = 0$ and thus $J_{i,j}(g_k, V_l) = 0$. Otherwise, let $\check{G}_l$ be the differential refinement of a free generator. Then from (3.6), we have $\int V_l = \int G_l$, where $G_l$ is the integral cochain $I(\check{G}_l)$. Therefore, $\int V_l$ has integral periods and $V_l$ is closed. As $I(\check{g}_k) = g_k$ is also closed and integral, we have

$$J_{i,j}(g_k, V_l) = \int g_k \cup V_l = \int [g_k] \cup [V_l] \in \mathbb{Z}. \tag{3.23}$$

This is a crucial aspect of our derivation; consider the following terms from (3.20):

$$\sum J_{i,j}(g_k, V_l)\delta n_{i-k-1} \cup m_{j-l-1} \tag{3.24}$$

This is supposed to be of the form $\delta\mathcal{L}_{\text{Sym}}$, like all of the other terms of (3.20), but this one is not and so should not contribute to $\mathcal{L}_{\text{Sym}}$. The corresponding term in the action is

$$S = 2\pi i \int_{\mathcal{N}_{d+2}} \sum J_{i,j}(g_k, V_l) \delta n_{i-k-1} \cup m_{j-l-1} \qquad (3.25)$$

where $n_{i-k-1} \in C^{i-k-1}(\mathcal{N}_{d+2}; \mathbb{Z}), m_{j-l-1} \in C^{j-l-1}(\mathcal{N}_{d+2}; \mathbb{Z})$ and so the integral of the cochain $\delta n \cup m$ is an integer. Since $J_{i,j}$ is also an integer the corresponding term of the action reduces to $2\pi i q$ with $q \in \mathbb{Z}$, so this is trivial in the path integral, and does not need to be included in $\mathcal{L}_{\text{Sym}}$. The remaining term $J_{i,j}\delta n \cup \delta B$ has a $B_{j-l-1} \in C^{j-l-1}(\mathcal{N}_{d+2}; \mathbb{R})$ factor, so we can choose to absorb $J_{i,j}$ into $B_{j-l-1}$, so long as $J_{i,j} \neq 0$.

Finally, we can write the remaining terms in (3.20) in the form $\delta\mathcal{L}_{\text{Sym}}$ and use Stokes' theorem to obtain the following action for the SymTFT

$$S = 2\pi i \sum_{\alpha,\gamma,k,l} \int_{\mathcal{M}_{d+1}} K_{i,j}(g_k, H_{l-1}) n_{i-k-1} \cup \delta m_{j-l-1} + J_{i,j}(g_k, V_l) n_{i-k-1} \cup \delta B_{j-l-2} \quad (3.26)$$

which is the main result stated in (1.2). Note that we have left the $\mathbb{R}$ superscript of $n_{i-k-1}^{\mathbb{R}}$ implicit in (3.26), and will continue to do so from now on.

The first term in (3.26) is the usual $BF$ term describing discrete higher-form symmetries (with $K_{i,j}$ the torsional linking pairing, see for instance [4, 42] for further discussion of this term), and the second term, involving $n_{i-k-1}^{\mathbb{R}} \in C^{i-k-1}(\mathcal{M}_{d+1}; \mathbb{R})$ and $B_{j-l-2} \in C^{j-l-2}(\mathcal{M}_{d+1}; \mathbb{R})$ a cochain description of the $U(1)/\mathbb{R}$ theory studied in [6, 7] for describing continuous abelian symmetries. To see this, we note that we can construct two basic operators out of $n_{i-k-1}$ and $B_{j-l-2}$: [3]

$$U_\alpha(\gamma) = e^{i\alpha \int_\gamma n_{i-k-1}}, \qquad (3.27a)$$

$$W_q(\gamma) = e^{2\pi i q \int_\gamma B_{j-l-2}}. \qquad (3.27b)$$

Invariance under large gauge transformations of $B_{j-l-2}$ forces $q \in \mathbb{Z}$, and due to the integrality of $n_{i-k-1}$ we have $\alpha \in \mathbb{R}/2\pi\mathbb{Z}$. This reproduces the operator content of the theories in [6, 7].

Finally, let us point out that our 11d bulk actions model $U(1)$ gauge fields, which implies by the analysis above that we are unable to reproduce SymTFTs with action $a \wedge db$, where both $a$ and $b$ are connections on $\mathbb{R}$ bundles. This is as expected, given that the theories that we are discussing are those with a string theory realisation, which are believed to always have spectra compatible with compact symmetry groups [68].

## 4 Kaluza-Klein reduction in terms of differential forms

The purpose of using a differential cohomology formulation of the bulk theory in section 3 was to allow us to derive the SymTFT for both discrete and continuous symmetries at the

---

[3]In [6, 7] the $U(1)$ gauge field takes values in $[0, 2\pi)$, whereas our $B_{j-l-2}$ fields are valued in $[0, 1)$, hence the factor of $2\pi$ in the exponent of our operator.

same time. However, if we are interested in just the continuous sector, then it is possible to perform Kaluza-Klein reduction with ordinary cohomology. In this section we perform such a reduction, as it might be illuminating for readers unfamiliar with the formalism used in the previous section. See [69] for how to instead compute SymTFTs for discrete symmetries in this way.

The full bulk Lagrangian in terms of differential cohomology is given in [32] as

$$-S = 2\pi i \int_{Y_{11}} \frac{i}{2e^2} R(\check{a}_i) \wedge *R(\check{a}_i) + \frac{i}{2m^2} R(\check{a}_j) \wedge *R(\check{a}_j) + \mathsf{h}(\check{a}_i \cdot \check{a}_j) \qquad (4.1)$$

where $i + j = 12$ as usual. Working at the level of differential forms we can write this as

$$-S = 2\pi i \int_{Y_{11}} \frac{i}{2e^2} db \wedge *db + \frac{i}{2m^2} dc \wedge *dc + b \wedge dc \,. \qquad (4.2)$$

If we expand the forms in terms of the harmonic forms of $L_{9-d}$ we get

$$b_p = \sum_{k=0}^{9-d} \sum_{i_k} f_{p-k}^{(i_k)} \wedge \omega_k^{(i_k)} \,, \qquad (4.3)$$

$$c_{10-p} = \sum_{k=0}^{9-d} \sum_{i_k} g_{10-p-k}^{(i_k)} \wedge \omega_k^{(i_k)} \,, \qquad (4.4)$$

where $\omega_k^{i_k}$ are the harmonic $k$-forms of $L_{9-d}$, and $f, g$ are forms on $\mathcal{N}_{d+2}$. Here $b_p$ corresponds to an $F_p$ field strength of Type II string theory that we are treating as a fundamental field of our bulk theory, and similarly $c_{10-p}$ corresponds to $*F_p$. To lighten our notation, below we will omit the index $i_k = 1, ..., h_k$, where $h_k$ is the $k$-th Betti number of $L_{9-d}$. Then,

$$db \wedge *db = \sum_k (df_{p-k} \wedge *_{\mathcal{N}} df_{p-k}) \wedge (\omega_k \wedge *_L \omega_k) \qquad (4.5)$$

and we define

$$\mathcal{K}_k = \int_{L_{9-d}} \omega_k \wedge *_L \omega_k \qquad (4.6)$$

such that the kinetic term of the $b$ form becomes

$$2\pi i \sum_k \mathcal{K}_k \int_{\mathcal{N}_{d+2}} \frac{i}{2e^2} df_{p-k} \wedge *df_{p-k}. \qquad (4.7)$$

This is the usual result that one would expect from a KK reduction of a Maxwell-like term, i.e. we just get multiple copies of Maxwell-like terms in the lower dimensional theory. We can find the dimensional reduction of the kinetic term for $c$ similarly. Finally, we consider the $BF$-like term:

$$b \wedge dc = \sum_{k=0}^{9-d} (f \wedge dg) \wedge (\omega_k \wedge \omega_{9-d-k}) \qquad (4.8)$$

and define the intersection numbers

$$\mathcal{J}_k = \int_{L_{9-d}} \omega_k \wedge \omega_{9-d-k} \qquad (4.9)$$

such that the $BF$ term becomes

$$2\pi i \sum_k \mathcal{J}_k \int_{\mathcal{N}_{d+2}} f \wedge dg \qquad (4.10)$$

where again we are summing over multiple $BF$ couplings.

Near the boundary we can write, by a choice of gauge transformation [32, 46],

$$f_i = e^{\alpha\tau} F_i \qquad (4.11)$$

and

$$g_i = e^{\alpha\tau} G_i \qquad (4.12)$$

where coordinates of $\mathcal{N}_{d+2}$ are given as $(x, \tau)$ with $x \in \mathcal{M}_{d+1}$. Here, $F_i$ is an 'electric' field strength with integral periods and $G_i$ is the corresponding 'magnetic' field strength, with both of these viewed as gauge parameters of the bulk $f_i, g_i$ fields, such that we can write the pure gauge $BF$ coupling as

$$\sum_k \mathcal{J}_k \int_{[-\epsilon,0]\times\mathcal{M}_{d+1}} e^{\alpha\tau} d(e^{\alpha\tau}) F_{p-k} \wedge G_{d+1-p+k} . \qquad (4.13)$$

From here we get the following action on the boundary

$$S_{\mathrm{Sym}} = 2\pi i \sum_k \mathcal{J}_k \int_{\mathcal{M}_{d+1}} F_{p-k} \wedge G_{d+1-p+k} . \qquad (4.14)$$

We can choose to write $F_{p-k} = dA_{p-k-1}$, with $A_{p-k-1}$ the electric gauge field, and then consider the following operators

$$W_q = e^{2\pi i q \int A} , \qquad (4.15a)$$

$$U_\alpha = e^{i\alpha \int G} . \qquad (4.15b)$$

For $W_q$ to be invariant under large gauge transformations of $A$, we require $q \in \mathbb{Z}$, and the fact that $G$ has integral periods implies that $\alpha \in U(1)$. Thus, this corresponds to a $U(1)^{(p-k-2)}$ symmetry of the QFT in the same way as we discussed below (3.27a) and (3.27b).

There is a puzzling asymmetry in this discussion, which was somewhat hidden in the discussion in the previous section, but which will reappear in the next section: why did we choose $A_{p-k-1}$ as our fundamental field when writing the holonomy operator, and not the connection $B_{d-p+k}$ whose field strength is $G$? This is certainly unnatural, since the bulk treated these fields on equal footing. To start to understand this, note that the operators (4.15), in the context of (generalised) Maxwell theory, are the ordinary Wilson line and the generator of the electric symmetry measuring Wilson lines, this last one written in terms of the magnetic field strength, using the electromagnetic duality relation $*F = G$ (coming from the equations of motion in the bulk [32, 46]). Reducing the parent type II string theory on a space with harmonic forms will lead to a Maxwell theory with massless $U(1)$ factors, so we certainly expect such operators to be present in the theory. But we

should also expect the dual 't Hooft and magnetic symmetry generators. Crucially, as we well know from Maxwell theory, these operators are *all* present in the theory of a *single* $U(1)$ propagating field, but their presentation depends on the electromagnetic duality frame that we choose when formulating the theory.

There is in fact a very analogous situation in the context of holography: consider the reduction of IIB supergravity on $\text{AdS}_5 \times X^5$, with $H^3(X^5; \mathbb{R}) = H^2(X^5; \mathbb{R}) = \mathbb{R}$, so there is a harmonic 2-form $\omega_2$ and a harmonic 3-form $\omega_3$ in the internal space $X^5$. A simple geometry with this property is the base of the conifold, $X^5 = T^{1,1} \coloneqq (SU(2) \times SU(2))/U(1)$ [70]. Writing $C_4 = A_1 \wedge \omega_3 + B_2 \wedge \omega_2$ would lead to the conclusion that there is a massless 1-form and a massless 2-form in $\text{AdS}_5$, but this argument needs to be supplemented by the self-duality condition $F_5 = *F_5$, which implies $dA_1 = *dB_2$, so these are not independent local degrees of freedom. The situation is rather that there is a single dynamical $U(1)$ field, and various duality frames in which to describe it.

Associate, to each duality frame, a boundary condition where the fundamental gauge connection in that duality frame has Dirichlet boundary conditions at infinity. Bulk electromagnetic duality does not in general respect this choice (it typically maps Dirichlet to Neumann), so these boundary conditions lead to distinct holographic dual theories on the boundary. We can rephrase this by saying that electromagnetic duality induces an action on the space of dual CFTs, via its action on the space of boundary conditions. This action of bulk electromagnetic duality on the space of CFTs was discussed in [71, 72], where it was shown to reproduce the effect of gauging global symmetries on the boundary CFT.

Coming back to the SymTFT context, we will now show in an example that a similar picture arises naturally from the string theory descent procedure once we formulate things in their natural K-theory setting: we will obtain SymTFTs where we see both the bulk $U(1)$ field and its electromagnetic dual on an equal footing.

## 5    6d $\mathcal{N} = (1,1)$ $\mathfrak{su}(N)$ theory

We have just seen how to derive the SymTFT for both continuous and discrete higher-form symmetries. We now illustrate these ideas in an explicit example, the 6d $\mathcal{N} = (1,1)$ $\mathfrak{su}(N)$ theory, whose SymTFT can be obtained by considering the link $L_3 = S^3/\mathbb{Z}_N$ on Type IIA string theory. We wrote down what the uplifted 11d bulk action should be for Type IIA in (2.11), and so we are considering the cases where we have $(i,j) = (3,9),(5,7)$. The cohomology of $L_3$ is given by

$$H^\bullet(L_3; \mathbb{Z}) = \{\mathbb{Z}, 0, \mathbb{Z}_N, \mathbb{Z}\} \tag{5.1}$$

and so we should consider (the differential uplift of) two generators: $\check{f}_3, \check{t}_2$. The zero-degree generator does not contribute to the defect group here, as one can calculate that $\frac{H_1(X_4, L_3)}{H_1(X_4)} = 0$. We have from [4, 32] that $\check{t}_2 = (t_2, \varphi_1, 0)$, such that $t_2 = -d\varphi_1$ as in (3.3), and

$$\int_{L_3} t_2 \cup \varphi_1 = \frac{1}{N} \bmod 1 \,. \tag{5.2}$$

Then, by following the discussion of section 3, we have

$$\check{f}_3 = (f_3, h_2, v_3) \tag{5.3}$$

with $f_3 = \text{vol}(L_3)$ [4]. Therefore, from (3.6) we have

$$\int_{L_3} f_3 = \int_{L_3} v_3 = 1 . \tag{5.4}$$

As discussed in section 2.1, the gauge non-invariance on the boundary of the bulk theory is captured by the terms

$$2\pi i \mathsf{h} \left( \int d\check{b}_2 \cdot d\check{b}_8 + d\check{b}_4 \cdot d\check{b}_6 \right) . \tag{5.5}$$

## 5.1 Symmetries from magnetic branes

Making a similar yet slightly restricted version of the ansatz in (2.15),

$$d\check{b}_2 = d\check{\beta}_2 + d\check{\beta}_0 \cdot \check{t}_2, \tag{5.6a}$$
$$d\check{b}_8 = d\check{\beta}_8 + d\check{\beta}_6 \cdot \check{t}_2 + d\check{\beta}_5 \cdot \check{f}_3, \tag{5.6b}$$
$$d\check{b}_4 = d\check{\beta}_4 + d\check{\beta}_2 \cdot \check{t}_2, \tag{5.6c}$$
$$d\check{b}_6 = d\check{\beta}_6 + d\check{\beta}_4 \cdot \check{t}_2 + d\check{\beta}_3 \cdot \check{f}_3 \tag{5.6d}$$

where we note that we are purposefully not including a $d\check{\beta}_1 \cdot \check{f}_3$ term in (5.6c), which will correspond to wrapping a $D2$-brane around a 3-cycle, giving a continuous $(-1)$-form symmetry. We will discuss this term separately below. We do not include a similar term in (5.6a) as this would correspond to wrapping a $D0$ brane around a 3-cycle, giving a $(-3)$-form symmetry, which, at least in this formalism, is not possible by dimensional reasons. (See [73] for an interpretation of such symmetries in a context similar to ours.) Therefore, by considering (1.2), we have that possible values to sum over are $k = 0, 2$, and $l = 0, 2, 3$, where there are no $\alpha, \gamma$ indices required as each degree of cohomology of $L_3$ has just one generator. We denote for $\check{\beta}$ that do *not* couple to the generator $\check{f}_3$

$$\check{\beta} = (n, A, 0) \tag{5.7}$$

and for $\check{\beta}$ that *do* couple to $\check{f}_3$

$$\check{\beta} = (m, B, 0). \tag{5.8}$$

Finally, for those $\check{\beta}$ that couple to $\check{t}_2$ we denote

$$\check{\beta} = (\tilde{n}, C, 0). \tag{5.9}$$

Note that this is slightly different to our notation in section 3, where we had the 'electric' gauge parameter $\check{\beta}_{i-k-1} = (n, A, 0)$ and 'magnetic' gauge parameter $\check{\beta}_{j-l-1} = (m, B, 0)$. By computing $K_{i,j}(g_k, H_{l-1}), J_{i,j}(g_k, V_l)$ for all values of $i, j, k, l$ mentioned above, using (5.2) and (5.4) and by considering dimensionality of the integrals, e.g. $\int_{L_3} c_n = 0$ if $n \neq 3$

for some cochain $c_n$, we are able to derive the whole $BF$ sector of the SymTFT for the 6d $(1,1)$ theory: the surviving $K_{i,j}$ are

$$K_{3,9}(t_2, \varphi_1) = K_{5,7}(t_2, \varphi_1) = \frac{1}{N} \tag{5.10}$$

and the surviving $J_{i,j}$ are

$$J_{3,9}(1, v_3) = J_{5,7}(1, v_3) = 1 \tag{5.11}$$

such that the resulting $BF$ sector of the SymTFT is

$$S^m_{(1,1)} = \frac{2\pi i}{N} \int \tilde{n}_0 \cup \delta \tilde{n}_6 + \tilde{n}_2 \cup \delta \tilde{n}_4 + 2\pi i \int n_2 \cup \delta B_4 + n_4 \cup \delta B_2 \tag{5.12}$$

where the first integral describes the discrete higher-form symmetries $\mathbb{Z}_N^{(-1)} \times \mathbb{Z}_N^{(5)}$ and $\mathbb{Z}_N^{(1)} \times \mathbb{Z}_N^{(3)}$ respectively, and the second integral describes the continuous higher-form symmetries $U(1)^{(3)}$, $U(1)^{(1)}$, respectively. Note that the $B_{p+1}$ terms in the second integral, after a sandwich, are background gauge fields for a $U(1)^{(p)}$ symmetry, as in [6, 7]. The $(d-p-1)$-dimensional symmetry operators are then generated by the $n_{d-p-1}$ cochains.

We can consider the form of the defect group given in [67] to see that these are in fact higher-form symmetries of the theory:

$$\mathbb{D} = \bigoplus_n \mathbb{D}^{(n)} = \bigoplus_{n=p-k} \frac{H_{k+1}(X_4, L_3)}{H_{k+1}(X_4)} = \bigoplus_{n=p-k} H_k(L_3; \mathbb{Z}) \tag{5.13}$$

where $p$ is the dimension of the $p$-branes that wrap the non-compact $(k+1)$-cycles of $X_4 = \mathbb{C}^2/\mathbb{Z}_N$ given by elements of $H_k(L_3; \mathbb{Z})$. In this last equality, we have used that there is an isomorphism between the non-compact $(k+1)$-cycles of $X_4$ and the compact $k$-cycles of $L_3$ in this example. By considering the long exact sequence of relative homology, as is discussed in [66, 67], the values of $k$ that we can consider here are $k = 1, 3$. Type IIA has $Dp$-branes given in electric-magnetic dual pairs as $(D0, D6), (D2, D4)$, and so the defect group from the equation above is then

$$\mathbb{D} = (\mathbb{Z}_N^{(1)} \oplus \mathbb{Z}^{(-1)})_{D2} \oplus (\mathbb{Z}_N^{(3)} \oplus \mathbb{Z}^{(1)})_{D4} \oplus (\mathbb{Z}_N^{(-1)})_{D0} \oplus (\mathbb{Z}_N^{(5)} \oplus \mathbb{Z}^{(3)})_{D6} \tag{5.14}$$

where $\mathbb{Z}^{(p)}$ defects correspond to a $U(1)^{(p)}$ symmetry, and $\mathbb{Z}_N^{(p)}$ defects correspond to a $\mathbb{Z}_N^{(p)}$ symmetry. Therefore, we can see that this is almost exactly the defect group we would expect from the SymTFT in (5.12). However, we have no term for the $U(1)^{(-1)}$ symmetry coming from the electric $(\mathbb{Z}^{(-1)})_{D2}$. We explicitly chose to not include a term in our ansatz corresponding to this earlier on. If we add the missing $d\check{\beta}_1 \cdot \check{f}_3$ term to our ansatz for $d\check{b}_4$ we get the following additional term for the 6d $(1,1)$ theory:

$$S^m_{(1,1)} \to S^m_{(1,1)} + 2\pi i \int m_1 \cup \delta A_5 \tag{5.15}$$

where the new term comes from the cross term $(d\check{\beta}_1 \cdot \check{f}_3) \cdot d\check{\beta}_6$. This is not the term that one would expect for a $(-1)$-form symmetry, it is rather the term one expects for a 4-form

symmetry. So our ansatz in its current form can capture the symmetries coming from the $D(9-i)$ branes, but it misses symmetries coming from electric $D(i-3)$ branes.

We will now show how a slightly different choice of bulk action can describe the symmetries from $D(i-3)$ branes, at the cost of missing the symmetries coming from $D(9-i)$ branes. In section 6 we will show that differential K-theory provides a unified formulation that can describe all expected symmetries.

## 5.2 Symmetries from electric branes

As a second attempt, let us suppose we replace the

$$2\pi i \mathsf{h}\left(\int \check{a}_5 \cdot \check{a}_7\right) \tag{5.16}$$

term in our 11d action (2.11) with

$$2\pi i \mathsf{h}\left(\int \check{a}_7 \cdot \check{a}_5\right). \tag{5.17}$$

The product of differential cochains is not (graded) commutative, so despite the superficial similarity this change can lead to potential changes in the SymTFT resulting from descent. We make the ansatz

$$d\check{b}_4 = d\check{\beta}_4 + d\check{\beta}_2 \cdot \check{t}_2 + d\check{\beta}_1 \cdot \check{f}_3\,, \tag{5.18}$$

$$d\check{b}_6 = d\check{\beta}_6 + d\check{\beta}_4 \cdot \check{t}_2 + d\check{\beta}_3 \cdot \check{f}_3\,. \tag{5.19}$$

We will use the same notation for the $\check{\beta}$ fields as in section 5.1. Computing the SymTFT coming from the new action, with the bulk $\check{a}_5, \check{a}_7$ fields as in (5.17), we get

$$S^e_{(D2,D4)} = \frac{2\pi i}{N}\int \delta\tilde{n}_2 \cup \tilde{n}_4 + 2\pi i \int m_3 \cup \delta A_3 + 2\pi i \int n_6 \cup \delta B_0 \tag{5.20}$$

where the final integral looks exactly like what we expect for a $U(1)^{(-1)}$ symmetry, and the sector corresponding to the discrete higher-form symmetries from $(D2, D4)$ branes is preserved, up to an integral by parts. The second term corresponds to a $U(1)^{(2)}$ symmetry, electromagnetic dual in the bulk to the $U(1)^{(1)}$ symmetry written in (5.14).

## 5.3 The $(2,0)$ theory

Let us briefly consider another related example, the $d = 6$ $\mathcal{N} = (2,0)$ $\mathfrak{su}(N)$ theory, engineered from IIB string theory on the same geometry as the previous example, $X_4 = \mathbb{C}^2/\mathbb{Z}_N$, with $L_3 = S^3/\mathbb{Z}_N$. The 11d bulk action we are considering here is given by (2.12), and so for (1.2) we need to sum over $(i,j) = (2,10), (4,8), (6,6)$. For now, we only wish to discuss the latter term, corresponding to the self-dual $F_5$ field strength. Making the most general ansatz for $d\check{b}_5$, we have

$$d\check{b}_5 = d\check{\beta}_5 + d\check{\beta}_3 \cdot \check{t}_2 + d\check{\beta}_2 \cdot \check{f}_3\,. \tag{5.21}$$

Considering all the allowed values of $k, l$, we can calculate the surviving $J_{i,j}, K_{i,j}$ and the resulting SymTFT that we get is

$$S_{D3} = \frac{2\pi i}{N} \int \tilde{n}_3 \cup \delta\tilde{n}_3 + 2\pi i \int n_5 \cup \delta B_1 + m_2 \cup \delta A_4. \qquad (5.22)$$

Again, we can compare this with the calculation of the defect group. For IIB, we have $Dp$-branes given in electric-magnetic dual pairs $(D(-1), D7), (D1, D5), (D3)$ where the $D3$-brane is self-dual. By using (5.13) for just $p = 3$ and the same values of $k$ as the 6d $\mathcal{N} = (1,1)$ example, we get

$$\mathbb{D}_{D3} = (\mathbb{Z}_N^{(2)} \oplus \mathbb{Z}^{(0)})_{D3}. \qquad (5.23)$$

We can see that the first integral of (5.22) accounts for the $\mathbb{Z}_N^{(2)}$ symmetry, where we note that for 6d theories $\mathcal{T}$ we have the property that $\mathcal{T} \cong \mathcal{T}/\mathbb{Z}_N^{(2)}$, i.e. gauging a discrete 2-form symmetry in 6d leaves the theory invariant. Then, the first term of the second integral in (5.22) corresponds to the $\mathbb{Z}^{(0)}$ defects, and we have an extra term coming from the $(d\check{\beta}_2 \cdot \check{f}_3) \cdot d\check{\beta}_5$ term of the 11d action. For the 6d $\mathcal{N} = (1,1)$ theory, and generally theories engineered in IIA, we were able to avoid terms that weren't described by (5.13) by considering the electric-magnetic dual branes separately, by swapping the order of the product of $\check{a}_i$ and $\check{a}_j$ in the action and making a different ansatz. However, this method doesn't work for theories engineered from Type IIB, as the $D3$ brane is self-dual and we cannot make two different ansatzes for the same field.

# 6 $\check{K}$-theory refinement

In our discussion so far we have seen that the SymTFT obtained from descent does depend on the precise ordering of the differential cochains in the bulk action. The dependence is somewhat mild: different orderings lead to different electromagnetic dual fields being represented in the SymTFT explicitly. Depending on whether we want to make the electric or magnetic representative manifest we could choose one form or the other. As discussed at the end of section 4, this question is ultimately related to whether we are gauging the continuous symmetries in the field theory or not.

Rather than saying that the bulk of the SymTFT depends on the choice of whether we gauge the symmetry or not, we take a different perspective, following what happens in holography [71, 72]: we now show that using a better formulation of RR fields in string theory, namely differential K-theory, leads to a universal SymTFT where both possibilities appear, linked by a duality relation.

## 6.1 $\check{K}$-theory bulk action

So far we have argued that one can obtain the $BF$ sector of the SymTFT for discrete and continuous symmetries by using ordinary differential cohomology, but in fact a formulation of RR fluxes in terms of differential $K$-theory is more natural from the string theory point

of view [49].[4] In this section and the next we will show how the $K$-theory formulation leads to an action which treats the two possibilities for the continuous SymTFT sector democratically. We start in this section with a quick review of the ideas of Belov and Moore [44, 45], and their further development in [32], and we refer the reader to those papers for more extensive discussion.

In order to understand the type IIA bulk action from an eleven dimensional perspective, let $\check{a} \in \check{K}^1(Y_{11})$ be

$$\check{a} = \check{a}_1 + \check{a}_3 + \check{a}_5 + \check{a}_7 + \check{a}_9 + \check{a}_{11} \tag{6.1}$$

with the group of differential $K$-theory cochains defined as in [65] (the details of the construction will not be essential for the point we want to make), and

$$\check{a}^* = \check{a}_1 - \check{a}_3 + \check{a}_5 - \check{a}_7 + \check{a}_9 - \check{a}_{11} \, . \tag{6.2}$$

For Type IIB, we can similarly define an $\check{a} \in \check{K}^0(Y_{11})$ as

$$\check{a} = \check{a}_0 + \check{a}_2 + \check{a}_4 + \check{a}_6 + \check{a}_8 + \check{a}_{10} + \check{a}_{12} \tag{6.3}$$

with $\check{a}^*$ defined in an alternating way, similarly to (6.2).

The 11d bulk action describing the RR sector is [32, 44, 45]

$$S = 2\pi i \mathsf{h} \left( \frac{1}{2} \int_{Y_{11}} \check{a} \cdot \check{a}^* \right) \, , \tag{6.4}$$

where the schematic factor of $1/2$ should be understood as a choice of quadratic refinement. Only terms of the form $\check{a}_i \cdot \check{a}_j$ with $i + j = 12$ appear in the resulting action, so we get

$$S = 2\pi i \mathsf{h} \left( \frac{1}{2} \int_{Y_{11}} \check{a}_i \cdot \check{a}_j - \check{a}_j \cdot \check{a}_i \right) \tag{6.5}$$

for $i + j = 12$.

## 6.2 Duality symmetric formulation for the SymTFT

We will now show how $\check{K}$-theory actions of the form (6.5) lead, by symmetry descent, to SymTFTs where symmetries from electric and magnetic branes appear simultaneously. In particular, the terms relevant to the SymTFT for the $F_4$ field strength term of IIA for the 6d $\mathcal{N} = (1, 1)$ are

$$S = 2\pi i \mathsf{h} \left( \frac{1}{2} \int_{Y_{11}} \check{a}_5 \cdot \check{a}_7 - \check{a}_7 \cdot \check{a}_5 \right) \, . \tag{6.6}$$

Making the most general ansatz for the gauge transformations

$$d\check{b}_4 = d\check{\beta}_4 + d\check{\beta}_2 \cdot \check{t}_2 + d\check{\beta}_1 \cdot \check{f}_3, \tag{6.7}$$

$$d\check{b}_6 = d\check{\beta}_6 + d\check{\beta}_4 \cdot \check{t}_2 + d\check{\beta}_3 \cdot \check{f}_3 \tag{6.8}$$

---

[4]The $K$-theory formulation is known to be limited, in that it is not known how to extend it to include the NSNS sector in a way that makes the dualities of string theory manifest, see [74] for a discussion of the relevant issues.

leads to the SymTFT

$$S_{(D2,D4)} = S_{\text{discrete}} + 2\pi i \int n_4 \cup \delta B_2 + n_6 \cup \delta B_0 + m_1 \cup \delta A_5 + m_3 \cup \delta A_3 \qquad (6.9)$$

where $S_{\text{discrete}}$ is just the first integral of (5.12), which we have already checked is under control. We see that the $K$-theory formulation exhibits both the $U(1)^{(-1)}$ symmetry coming from a $D2$ brane and the $U(1)^{(1)}$ symmetry coming from a $D4$ brane simultaneously, corresponding to the first two terms of the integral. However, both electromagnetic dual terms also appear, corresponding to the two final terms of the integral. These $U(1)$ fields are, of course, still linked by electromagnetic duality, so this formulation is symmetric at the cost of being redundant.

## 6.3 The anomaly sector

We now would like to argue, extending the discussion in [32], that including an $H$-twisting of our $\check{K}$-theoretical action allows us to compute anomaly terms of the SymTFT. We will use the 6d $(1,1)$ theory as a guiding example. Let us consider the $\check{K}_H$ formulation of our bulk action

$$S = 2\pi i \mathsf{h} \left( \int_{Y_{11}} \frac{1}{2} \check{a} \cdot \check{a}^* \right) - 2\pi i \mathsf{h} \left( \int_{W_{12}} \check{a} \cdot \check{H} \cdot \check{a}^* \right) \qquad (6.10)$$

where $W_{12} = \mathcal{N}_8 \times X_4$. We have that the integral over $L_3$ of the gauge transformations of $\check{a}$ of this term should give us $\mathcal{L}_{\text{anomaly}}$ when we restrict $\mathcal{N}_{d+2}$ back to $\mathcal{M}_{d+1}$. Here, $\check{H}$ is a differential refinement of the NSNS 2-form field $B_2$, and we can choose $\check{H} \in \check{H}^3(\mathcal{N}_8)$ such that we implicitly pullback $\check{H}$ to $W_{12}$, as usual. Then, let $\check{H} = (g, \gamma, \Gamma)$, such that $\delta\gamma = \Gamma - g$. We can consider the terms

$$\check{a}_i \cdot \check{H} \cdot \check{a}_j \qquad (6.11)$$

as composing our 12d Lagrangian, where now $i + j = 10$. Then, under a gauge transformation:

$$S_H = 2\pi i \mathsf{h} \left( \int_{\mathcal{N}_{d+2} \times X_{10-d}} \check{a} \cdot \check{H} \cdot \check{a}^* \right) \rightarrow 2\pi i \mathsf{h} \left( \int_{\mathcal{N}_{d+2} \times X_{10-d}} d\check{b} \cdot \check{H} \cdot d\check{b}^* \right) \qquad (6.12)$$

where now we can use Stokes' theorem to obtain

$$\Delta S_H = -2\pi i \mathsf{h} \left( \int_{\mathcal{N}_{d+2} \times L_{9-d}} \check{b} \cdot \check{H} \cdot d\check{b}^* \right) + 2\pi i \mathsf{h} \left( d \int_{\mathcal{N}_{d+2} \times X_{10-d}} \check{b} \cdot \check{H} \cdot d\check{b}^* \right) \qquad (6.13)$$

where in this note we will not consider the contribution from this second term, as is done in [32].

Considering 6d (1,1) theory as an example, we can choose the same most general ansatz[5] as in section 5.2 for $\check{b}_{i-1}, \check{b}_{j-1}$, and we have that the terms contributing to $\mathcal{L}_{\text{anomaly}}$ from our 12d Lagrangian will be

$$\check{\beta}_{i-1} \cdot \check{H} \cdot (d\check{\beta}_{j-4} \cdot \check{f}_3) + (\check{\beta}_{i-4} \cdot \check{f}_3) \cdot \check{H} \cdot d\check{\beta}_{j-1} \qquad (6.14)$$

---

[5]We consider only free generators here, as anomalies of discrete symmetries from torsion generators has already been discussed in [32].

where we sum over both $(i, j)$ and $(j, i)$ due to the $K$-theoretic formulation of the action. Integrating over $L_3$, we get

$$\Delta S_H = 2\pi i \int_{\mathcal{N}_8} n_{i-1} \cup g_3 \cup \delta B_{j-5} + m_{i-4} \cup g_3 \cup \delta A_{j-2} \tag{6.15}$$

$$= 2\pi i \int_{\mathcal{M}_7} n_{i-1} \cup g_3 \cup B_{j-5} + m_{i-4} \cup g_3 \cup A_{j-2} \tag{6.16}$$

$$- 2\pi i \int_{\mathcal{N}_8} \delta n_{i-1} \cup g_3 \cup B_{j-5} + \delta m_{i-4} \cup g_3 \cup A_{j-2} \tag{6.17}$$

such that we have

$$S_{\text{anomaly}} = 2\pi i \sum_{i+j=10} \int_{\mathcal{M}_7} n_{i-1} \cup g_3 \cup B_{j-5} + m_{i-4} \cup g_3 \cup A_{j-2}. \tag{6.18}$$

Therefore, for the 6d (1,1) theory we can write the anomaly terms between continuous symmetries as

$$S_{\text{anomaly}}^{(1,1)} = 2\pi i \int_{\mathcal{M}_7} n_0 \cup g_3 \cup B_4 + n_2 \cup g_3 \cup B_2 + n_4 \cup g_3 \cup B_0 + m_1 \cup g_3 \cup A_3 + m_3 \cup g_3 \cup A_1. \tag{6.19}$$

Slightly extending the computation of anomaly terms given in [32], we can make another expansion of $\check{H}$ in the following way

$$\check{H} = \check{H}_0 \cdot \check{f}_3 \tag{6.20}$$

where $\check{H}_0 = (g_0, 0, \Gamma_0)$, which gives

$$\check{H} = (g_0 \cup f_3, g_0 \cup h_2, \Gamma_0 \wedge v_3). \tag{6.21}$$

Then, we require that $i + j = 10$ as usual for our $\check{a}_i \cdot \check{H} \cdot \check{a}_j$, but now we need the following coupling for this choice of $\check{H}$:

$$\check{\beta}_{i-1} \cdot (\check{H}_0 \cdot \check{f}_3) \cdot d\check{\beta}_{j-1}. \tag{6.22}$$

Evaluating this and integrating over $L_3$ to obtain the anomaly term as before, we get

$$S_{\text{anomaly}} = 2\pi i \sum_{i+j=10} \int_{\mathcal{M}_7} n_{i-1} \cup g_0 \cup A_{j-2}. \tag{6.23}$$

We can choose to set $g_0 = 1$. For example, the choice $(i, j) = (5, 5)$ gives the term

$$n_4 \cup g_0 \cup A_3 \tag{6.24}$$

which, when included in the SymTFT for the 6d (1,1) theory given in (6.9), we get equations of motion, after setting $g_0 = 1$,

$$- \delta B_2 = A_3 \tag{6.25}$$

$$- \delta m_3 = n_4. \tag{6.26}$$

Therefore we can see, for example, that picking Dirichlet boundary conditions for $A_3$ forces Neumann boundary conditions for $B_2$, such that we cannot choose both the $U(1)^{(1)}$ and dual $U(1)^{(2)}$ symmetries at the same time. Additionally, if we include the anomaly terms from (6.19), we also find an anomaly between $B_2$ and $B_0$, where these are defects for $(\mathbb{Z}^{(1)})_{D4}, (\mathbb{Z}^{(-1)})_{D2}$ from (5.14) respectively, and give symmetries coming from electric-magnetic dual branes such that only one can be present in a given theory.

**Note added.** As we were finishing this note we were informed about the upcoming work [75], which has some overlap with our discussion. We thank the author for agreeing to coordinate submission.

## Acknowledgments

We thank Felix Christensen and Jamie Pearson for helpful discussions. I.G.E. thanks S. Hosseini for collaboration on [32]. F.G. is funded by the STFC grant ST/Y509334/1. I.G.E. is partially supported by STFC grants ST/T000708/1 and ST/X000591/1 and by the Simons Foundation collaboration grant 888990 on Global Categorical Symmetries.

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
