# Peer review of "SymTFTs for U(1) symmetries from descent"

_SciPost Physics_

## Round 1 · Referee Report · Anonymous (Referee 1) · 2025-8-9

Report

This paper extends the symmetry descent procedure to obtain SymTFTs for $U(1)$ higher-form symmetries of geometrically engineered QFTs in Type II string theory. The construction is carried out in the Hopkins–Singer formalism, making it possible to treat discrete and continuous sectors in a unified way. A complementary derivation using Kaluza–Klein reduction in terms of differential forms provides a more direct physical picture of the continuous sector, while a refinement to differential K-theory gives a formulation in which electric and magnetic brane sectors appear on equal footing. The general discussion is illustrated with the 6d $\mathcal{N}=(1,1)$ and $\mathcal{N}=(2,0)$ $su(N)$ theories, where the defect group and anomaly structure follow directly from the geometry.

The derivations are technically sound and the results match and extend existing SymTFT constructions. Some steps are a bit dense and could be expanded for clarity, but overall the paper gives a coherent and useful extension of the SymTFT framework for continuous symmetries and will be of interest to those working on generalized symmetries, SymTFTs, and their string theory realizations.

The paper can be published to SciPost after the following minor points are addressed or answered.

  1. In introduction equation (1.2) the symbols $g_k$, $H_{l-1}$, $V_l$ appear to be undefined. I suggest either removing or provide an explanation.

2.On page 6, I am not sure that the terminology "gauging U(1) with discrete topology" is correct. This should either be trivial or correspond to gauging a discrete subgroup of U(1). The more accurate term would be "flat gauging the U(1)".

  1. On page 7, is $\hat Z^k$ defined anywhere?

  2. In subsection 6.1 the discussion involves differential K-theory, but it seems that the main requirement is the democratic formulation of type II supergravity. It would be helpful to either add more detail after equation (6.1) or emphasize that what is actually used is the democratic formulation only.

  3. The last paragraph on page 20 would benefit from a more detailed analysis of the boundary conditions compatibly with the equations of motion. As it currently stands, the anomaly condition that determines the choice of electromagnetic frame feels somewhat cryptic and not very explicit.

Recommendation

Publish (easily meets expectations and criteria for this Journal; among top 50%)

---

## Round 1 · Referee Report · Anonymous (Referee 2) · 2025-8-12

Strengths

1) The paper offers, in section 2, a useful and streamlined summary of several interesting results available in the literature in different places.

2) The paper fills an important gap: it successfully derives the SymTFT for continuous abelian symmetries from string theory.

3) The paper provides concrete examples that help the reader in understanding the main technique used.

Weaknesses

1) The context (e.g. the assumption, the dimensions, the goal) is not always clear. In section 3 for instance, that presents the main result, it is not clearly stated what is the starting point of the derivation, which theory lives in which dimension and so on.

2) In certain points the presentation is quite chaotic, and the main ideas are often hidden behind the formalism.

3) The notation is quite heavy and not always consistent. In particular certain symbols have different meaning in different places.

Report

The paper shows how to obtain the SymTFT for abelian symmetries, including finite and, notably, continuous symmetries, for QFTs with a string theory construction. In particular, the authors consider type IIA/IIB string theory compactified on X_d times a CY cone C(L), with L a compact manifold. This produces an SCFT in the remaining directions X_d. Following the same logic as in previous works (2112.02092, 2404.16028), the authors derive the SymTFT for the theory living on X_d by reducing the topological term of the SUGRA theory on the compact manifold L. The main novelty with respect to previous works is the inclusion of non-torsional cycles of L, which give rise to continuous symmetries. This is expected, as wrapping branes on these cycles and the radial direction produces defects with Z-valued charge. This is confirmed in this work by explicitly performing the dimensional reduction.

While the treatment of non-torsional cycles is, at least in principle, amenable to standard techniques in differential geometry, the authors prefer to employ the formalism of Hopkins-Singer differential cochains. This is necessary when dealing with torsional cycles; hence the formalism has the advantage of providing a unified derivation of both finite and continuous symmetries. It has, however, the disadvantage of making the presentation much more technical and hard to follow for any reader who is not familiar with the Hopkins-Singer formalism. In section 4, a derivation with standard KK reduction is provided, but the presentation is not fully independent, as the logic is explained in section 3, and, more importantly, the authors do not fully motivate the reader to study the heavier formalism necessary to read the other sections. It would be very helpful if the authors could provide at least one clear reason why formulating continuous symmetries in the Hopkins-Singer language is a useful approach. It seems to me that there is one: the existence of higher-group structures where a 0-form U(1) symmetry is mixed, through a non-trivial Postnikov class, with a finite higher-form symmetry. I think that, in those cases, to capture the SymTFT terms responsible for the higher-group, one is forced to treat the two sets of symmetries within the same formalism (hence Hopkins-Singer). Providing such an example would really motivate the reader to study the paper and would increase the importance of this work.

Moreover, in my opinion, the clarity of the presentation would benefit from switching sections 3 and 4: presenting the main ideas and derivation first in a context that is more familiar to the average reader, and then moving to the more technical (yet well-motivated) section would make the paper much smoother to read.

The final section 6 provides a very interesting new observation. By using the K-theory formalism, it is argued that one does not need to change the SymTFT if the U(1) is gauged dynamically (as was previously explained in 2401.10165), but that the dynamical gauging procedure is obtained simply by changing the boundary conditions. It would be nice if the authors could provide an intuitive explanation for why the K-theory language allows this nice fact, which does not seem to be possible using differential cohomology.

I also have a few questions and comments on specific points (some of them will also appear in the "requested changes"): 1) Below eq. 1.2, when it is stated that K_{ij} is rational, it is worth mentioning that the denominator is related to the order of certain homology groups of L.

2) The logic in the discussion from eq. 2.8 is somewhat chaotic, as explained. In the usual procedure, one puts type II string theory on M_{10} = M_d x C(L_{9-d}) and reduces on L_{9-d}, obtaining a TQFT on M_{d+1} = M_d x [0,1]. Here, on the other hand, string theory lives in M_{d+1} x L_{9-d}, which is not equivalent to M_{10} (in particular, it is not singular), and the former is realized as the boundary of Y_{11} = N_{d+2} x L_{9-d}. Could the authors clarify why the two procedures are supposed to be equivalent?

3) Why is the first term in eq. 3.1 there? Isn’t that included in the sum for k=0?

4)I appreciate that, from eq. 3.14 onward, the authors avoid writing the index alpha (and also gamma below), but I think it is quite unhelpful for the reader to have formulas where the index being summed over does not appear at all in the summand. For instance, equations 3.18 and 3.19 have sums over k, l, alpha, gamma, but none of them appear, while k appears again in 3.20.

5) There is something unclear in the way n starts appearing from eq. 3.13 up to eq. 3.20, and then in eq. 3.26. It seems to me that the various n's in the final result all have the same origin (eq. 3.13), but consistency seems to suggest that they should be different objects. In particular, as implied by the comment below 3.27, the n in the second term of 3.26 does not have any large gauge transformations that would quantize alpha (unlike B). However, if I understand correctly, the first term in 3.26 is supposed to describe finite symmetries; hence, the topological operators from that sector should have quantized charge.

6) What is the physical meaning of the (-1)-form symmetry of the (1,1) theories? Is there a theta angle in these theories?

7) In section 6, the authors argue that the K-theory formulation treats “democratically” a theory with a continuous symmetry and its gauged version: gauging U(1)^{(p)} in d dimensions leads to a dual (d-p-3)-form U(1) symmetry. E.g., gauging a U(1)^{(1)} in 6d produces a theory with U(1)^{(2)}. Clearly, the two symmetries cannot exist simultaneously; hence, the Dirichlet boundary for both must be forbidden. This is argued at the very end of section 6.3, where the inclusion of anomalies is discussed. However, this basic fact should be independent of the presence of anomalies and should already arise immediately after eq. 6.9. It is not clear to me why this is true, however.

Requested changes

1) The author should supplement the actual material with at least one example in which the necessity of the Hopkins-Singer formalism for continuous symmetries is clear and well-motivated. This can be a higher-group involving a continuous 0-form symmetry and a finite higher-form symmetry, or an instance of a mixed anomaly between the two.

2) Section 2.1 has essentially two logically distinct parts. The first is an introduction to the Hopkins-Singer formalism and is independent from the second part (starting around 2.8), which is in the context of string theory. I suggest clearly separating the two parts.

3) I think that before eq. 2.8 it is worth spending a few words on integration (even briefer than in 2404.16028), otherwise some readers unfamiliar with differential cohomology might be confused by integrating a 12-form on an 11d manifold.

4) Improve the clarity of the indices mentioned in point 4 of the report.

5) Addressing point 5 of the report: it would also be useful if the authors could clarify the operator content coming from the first term in 3.26.

6) Addressing point 7 of the report.

Recommendation

Ask for major revision

---

## Editorial Decision

awaiting_resubmission